# Combinations of Spent Grains as Sources of Valuable Compounds with Highly Valuable Functional and Microbial Properties

Mukul Kumar [1,*], Anisha Anisha [1], Deepika Kaushik [2], Jasjit Kaur [1], Shubham Shubham [3], Alexandru Vasile Rusu [4,†], João Miguel Rocha [5,6,7,*,†] and Monica Trif [8]

1 Department of Food Technology and Nutrition, Lovely Professional University, Phagwar 144411, India; anishaarora2302@gmail.com (A.A.); jassk0508@gmail.com (J.K.)

2 Department of Biotechnology, Faculty of Applied Sciences and Biotechnology, Shoolini University, Solan 173229, India; dk4275388@gmail.com

3 Department of Innovation Engineering University of Salento, 72100 Brindisi, Italy; shubhamkolish@gmail.com

4 CENCIRA Agrofood Research and Innovation Centre, 400650 Cluj-Napoca, Romania; rusu_alexandru@hotmail.com

5 Universidade Católica Portuguesa, CBQF Centro de Biotecnologia e Química Fina Laboratório Associado, Escola Superior de Biotecnologia, Rua Diogo Botelho 1327, 4169-005 Porto, Portugal

6 LEPABE—Laboratory for Process Engineering, Environment, Biotechnology and Energy, Faculty of Engineering, University of Porto, Rua Dr. Roberto Frias, s/n, 4200-465 Porto, Portugal

7 ALiCE—Associate Laboratory in Chemical Engineering, Faculty of Engineering, University of Porto, Rua Dr. Roberto Frias, s/n, 4200-465 Porto, Portugal

8 Food Research Department, Centre for Innovative Process Engineering (CENTIV) GmbH, 28857 Syke, Germany; monica_trif@hotmail.com

* Correspondence: mukulkolish@gmail.com (M.K.); jmfrocha@fe.up.pt (J.M.R.)

† These authors contributed equally to this work.

**Abstract:** The potential of spent grains as a source of valuable compounds with various properties has gained attention. They are the by-product of the brewing process, typically resulting from the beer-making process. Five different mixed combinations of spent grains of barley, wheat, rice, maize and finger-millet were formulated and further analyzed and compared. Barley and wheat (BW), barley and rice (BR), barley and maize (BM), and barley and finger-millets (BF) were mixed in a ratio of 1:1 (*w/w*) and ground into a fine powder to study their techno-functional, phytochemical and in vitro properties. The techno-functional, phytochemical and in vitro properties of barley and maize (BM) were found to be the best choice, making it a promising candidate for applications in value-added products. The WAI (water absorption index) of BM (5.03 g/g) was the highest compared to BB (3.20 g/g), BF (3.56 g/g), BR (4.10 g/g) and BW (4.33 g/g), whereas the WSI (water solubility index) and OAC (oil absorption capacity) of BM (7.06% and 1.90 g/g, respectively) were lower than BW (7.60% and 2.24 g/g, respectively), BR (8.20% and 2.30 g/g, respectively), BF (9.67% and 2.57 g/g, respectively) and BB (10.47% and 2.70 g/g, respectively). A higher percentage of inhibition of DPPH (44.14%) and high phenolic and flavonoid contents (72.39 mg GAE/gm and 66.03 mg QE/gm, respectively) were observed in BM. It also showed higher in vitro properties like amylase and lipase inhibition assay (89.05% and 62.34%, respectively) than the other combinations. The present study provides valuable information about the differences between spent grain varieties and their combinations, with potential applications in various industries.

**Keywords:** spent grain; formulation; health; in vitro; phytochemical

## 1. Introduction

Many attempts have been made to reuse agro-industrial by-products to reduce organic waste, to contribute towards environment conservation, to preserve bio-resources and to

manufacture value-added food products at low prices. Utilizing spent grains for their potential therapeutic properties is an interesting concept, and a great initiative under the sustainability context. Spent grains refer to the by-product of the brewing process, primarily consisting of barley, wheat or other grains used to extract sugars for fermentation. While spent grains are commonly used as animal feed or compost, they also contain bioactive compounds that have numerous therapeutic and nutraceutical properties. Brewer′s spent grain (BSG) is one of the major by-products generated during the beer brewing process. It is a significant source of waste in the brewing industry, but it also presents several potential applications and benefits. BSG mainly consists of the solid residue left after the sugars and other soluble components have been extracted from the malt during the mashing and lautering processes [1]. BSG is the brewing industry′s most prevalent by-product, contributing to about 85% of all by-products generated in the brewing industry [2]. The sustainable management of BSG is an important consideration for breweries, and finding effective and environmentally friendly ways to use or dispose of this by-product can contribute to reducing the waste and environmental footprint of the brewing industry. The type of barley, harvest period, malting, mashing circumstances, and brewing technology affects the composition of spent grains [3]. As a lingo-cellulosic material, it contains roughly 50 to 70% fibers, 15 to 25% proteins, 10 to 30% lipids, 1 to 12% starch and 2 to 5% ashes, among other ingredients [4]. It is made up of solid leftovers from the manufacturing of wort, such as barley husk and other fibrous materials. Barley malt grains, after the brewing process, contain three main parts: the husk, pericarp and seed coat. These parts are rich in cellulose, lignin, hemicellulose, lipids and proteins [5]. They also contain beneficial nutrients such as proteins and fibers, and secondary metabolites like antibiotics, mycotoxins, alkaloids, flavonoids and plant growth factors. The grains also have food-grade pigments, ferulic acid and p-coumaric acid [6]. Due to the large quantities produced, the low market value and the high moisture content, disposing of the grains has become a concern for the brewing industry. Landfill disposal or burning can cause environmental contamination, so proper disposal methods are important. [7]. BSG degrades quickly due to its high moisture content (>70% $w/w$). Therefore, the drying process can enhance the marketability and use of brewer′s spent grain. However, drying helps reduce product volume, lowering transportation and storage expenses [8]. The leftover quantity of BSG can be used for animal feed, fodder and food industries due to the higher nutritional value (fiber and protein content). Brewer′s spent grain can be considered as an energy substrate used for biogas production through anaerobic digestion, thus reducing the disposal costs, energy and carbon footprint [7,9]. By incorporating BSG into agri-food products, manufacturers can make them more nutritious and appealing to health-conscious consumers. Additionally, utilizing BSG as an ingredient in agri-food products can contribute to sustainability efforts by reducing waste and using a by-product that would otherwise be discarded [10]. Spent grains are now commonly utilized in various industries such as food, feed, fodder and composite production due to their affordability [11]. The current research study was conducted to assess the techno-functional, phytochemical and in vitro properties of different types of spent grains. This research effort emphasizes the potential of spent grains to produce healthy agri-food products.

## 2. Materials and Methodology

The grains were procured from the local market of Jalandhar, Punjab, India. The five different compositions of spent grains used for the analysis were barley, barley and wheat (1:1, $w/w$), barley and rice (1:1, $w/w$), barley and maize (1:1, $w/w$), and barley and finger-millets (1:1), which were encoded as BB, BW, BR, BM and BF, respectively. These samples (BB, BW, BR, BM and BF) were dried at 30°C in a hot air tray drier. The BSG powder was prepared through an electrical grinder (Sujata Powermatic plus 900 watts juicer mixer grinder) and sieved through a sieve shaker of different mesh sizes, viz. 0.425, 0.300, 0.150 and 0.075 mm, and the powder collected in the 0.075 mm sieve was used for further analysis.

## 2.1. Extraction of Brewer′s Spent Grains

Spent grains were obtained according to the method given by Jaeger et al. [12]. First, the grains were cleaned to remove the impurities like husk, dirt and foreign material. The grains were steeped in distilled water for two days under ambient conditions. The grains were then germinated for 6–7 days under ambient conditions after 4–6 weeks of dormancy, which increased the amount of enzymes in the grains. Afterwards, the grains were dried in a tray drier (PPI FiniX72, PPI Projects PVT. Limited, Delhi, India) at the temperature of 30 °C until their moisture content reached 4–5%. The grains were stored to obtain homogeneity and equilibrium for 3–4 weeks. After storage, the obtained product is known as malt. The temperature was then increased to 78 °C while the water and malt were combined to speed up the enzymatic hydrolysis of the malt. The process of converting enzymes is called mashing, and it results in a sweet liquid called wort. After fermentation, the filtered wort was used to make beer, while the solid residue was called "Brewer′s Spent Grains" (BSG). The BSG was then dried in a tray drier at 30 °C to turn it into a powder, which was used for further analysis.

## 2.2. Techno-Functional Properties

The method of Ndayishimiye et al. [13] was used to determine the bulk density (BD) and tapped density (TD) in g/mL. The following equation was used to calculate the bulk density using the BSG powder′s weight-to-volume ratio:

$$\text{BD (g/mL)} = \frac{Weight\ of\ the\ sample}{Volume\ of\ the\ sample} \tag{1}$$

The ratio of the BSG powder′s weight and final tapped volume was used to calculate the tapped density (TD). The equation is given below:

$$\text{TD (g/mL)} = \frac{Weight\ of\ the\ sample}{Final\ Tapped\ Volume} \tag{2}$$

Carr′s index (CI) and the Hausner ratio (HR) were estimated using the procedure of Hetclova et al. [14]. Carr′s index indicates the compressibility of the BSG powder and was determined through the following equation:

$$\text{CI (\%)} = 100 \times \frac{TD - BD}{TD} \tag{3}$$

The Hausner ratio (HR) expresses the flowability of the BSG powder and was determined by the equation:

$$\text{HR} = \frac{TD}{BD} \tag{4}$$

The angle of repose ($\varphi$) was determined by the method given by Segovia-Huarcaya et al. [15]. The following equation was used to calculate the angle of repose:

$$\varphi = \tan^{-1} \frac{2H}{D} \tag{5}$$

## 2.3. Water Absorption Index (WAI) and Water Solubility Index (WSI)

The Jiang et al. [16] method was used to determine the water absorption index (WAI) and water solubility index (WSI). Three grams of sample was suspended in deionized water at 30 °C for 30 min, stirred slightly every 5 min, and then centrifuged at 5000 rpm for 10 min. The supernatant liquid was transferred to the aluminum box and dried at 105 °C until constant weight. The residue was weighed, and water hydration parameters were calculated by the following formula.

$$\text{WAI (g/g)} = \frac{Weight\ of\ hydrated\ residue}{Dry\ weight\ of\ sample} \tag{6}$$

$$\text{WSI (\%)} = \frac{\text{Weight of dissolved solids in supernatant}}{\text{Dry weight sample}} \tag{7}$$

### 2.4. Swelling Capacity (SC)

The swelling capacity was calculated by the procedure given by Shroti et al. [17]. A tenth of a gram of sample was added in 40 mL of water and kept for 24 h, and the SC calculated by the equation calculated:

$$\text{SC (mL/g)} = \frac{Final\ volume\ (Vf)}{weight\ of\ sample} \tag{8}$$

whereas *Vf* is the final volume after 24 h.

### 2.5. Foaming Capacity (FC)

The foaming capacity (FC)was calculated by the procedure given by Vieira et al. [18]. Two grams of sample was homogenized for 1 min, and the foam volume was noted and used to calculate FC through the following equation:

$$\text{Foaming capacity (\%)} = \frac{Va - Vb}{Vb} \times 100 \tag{9}$$

where *Va* = volume of liquid + foam (mL), *Vb* = volume of mixture before blending.

### 2.6. Oil Absorption Capacity (OAC)

The oil absorption capacity (OAC) was determined according the procedure described by to Singh et al. [19]. Briefly, 0.3 g of sample was mixed in soybean oil (3 mL) and centrifuged for 30 min at 2060 rpm. Then, the residue was weighed and the OAC of spent grain powder was calculated using the equation:

$$\text{OAC (g/g)} = \frac{Weight\ of\ sample + oil}{Weight\ of\ sample} \tag{10}$$

### 2.7. Estimation of Antioxidant Efficacy

The DPPH (2,2-diphenyl-1-picrylhydrazyl) assay was used to measure the antioxidant activity. The ascorbic acid (reference) was used to plot the calibration curve. A tenth of a gram of sample was mixed in 10 mL of ethanol. Then, 250 μL of this solution was mixed with 2.0 mL of 0.1 mM DPPH ethanol solution and further incubated in the dark for 30 min. The decrease in the DPPH free radical was then determined by measuring the solution's absorbance at 517 nm with a spectrophotometer [20]. The percentage of DPPH scavenging activity was calculated using the following formula:

$$\% = \frac{\text{A} - \text{A1}}{\text{A}} \times 100 \tag{11}$$

where A = Absorbance of the control and A1 = Absorbance of the extract.

### 2.8. Determination of Total Phenol Content

The Folin–Ciocalteu (FC) assay was used to calculate the total phenolic content. 0.1 g of sample was mixed in 10 mL of ethanol, 250 μL of this solution was mixed with 10 μL of FC reagent, then 100 μL of sodium carbonate (7.5%) solution and incubated for 30 min. The absorbance at 765 nm was recorded using a spectrophotometer (Evolution™ One/One Plus UV-1800 Shimadzu UV Spectrophotometer, Shimadzu, Kyoto, Japan) [20].

### 2.9. Determination of Total Flavonoid Content

The method described by Merten et al. [21] was used to determine the total flavonoid content of spent grains. A tenth of a gram of sample was mixed in 10 mL of ethanol. Then, 250 μL of this solution was mixed with 10 μL of 5% sodium nitrite solution, followed by the addition of 10 μL of aluminum chloride (10%) solution after 5 min. Then, 100 μL of 1 M

NaOH solution was added after an incubation time of 6 min. The absorbance at 510 nm was recorded using a spectrophotometer (Evolution™ One/One Plus UV-1800 Shimadzu UV Spectrophotometer, Shimadzu, Kyoto, Japan).

### 2.10. Amylase and Lipase Inhibition Assay

The method of Kumar et al. [22] was used to evaluate the amylase inhibition activity, whereas the method described by Patil et al. [23] was employed to estimate the results of the lipase inhibition assay.

### 2.11. Glucose Uptake Assay

The glucose uptake assay was performed to determine the impact of glucose movement using the method of D'Souza et al. [24]. In brief, 0.5 g of the sample was mixed with 50 mL of different concentrations of anhydrous glucose (10–200 mmol/L) and incubated at 37 °C for 6 h. After centrifugation at 4500 rpm for 20 min at 25 °C, the amount of adsorbed glucose was estimated by quantifying the concentration of glucose in the supernatant using the glucose oxidase enzyme kit. CAG was expressed as mmol of glucose adsorbed per gram of sample (mmol/g).

### 2.12. Total Protein Content

The protein estimation was performed using the Kjeldahl method [25] and quantified as follows:

$$\text{Nitrogen (\%)} = \frac{[A - B] \times 0.0014 \times volume\ of\ digest}{Aliquot\ taken \times S} \times 100 \tag{12}$$

where $A$ is the sample titer (mL), $B$ is the blank titer (mL), and $S$ is the weight of the sample taken. The following formula estimated the protein content, where 6.25 is the conversion factor:

$$\text{Protein (\%)} = \text{Nitrogen (\%)} \times 6.25 \tag{13}$$

### 2.13. Crude Fiber (CF)

The crude fiber (CF) content estimation was performed using the method given by Madubuike et al. [26]. The CF (%) was calculated according to the following expression:

$$\text{CF (\%)} = \frac{(W2 - W1)}{(W1 - W)} \times 100 \tag{14}$$

where $W$ = empty crucible weight (g), $W1$ = weight of crucible + sample before ignition (g) and $W2$ = weight of crucible + sample after ignition (g).

### 2.14. Fourier Transform Infrared Spectroscopy (FTIR)

Functional groups in spent grain powder were evaluated and spectra were recorded using FTIR analysis (PerkinElmer FTIR spectrophotometer, equipped with KBr beam splitter) with 5 mg of sample and 5 mg KBr for the qualitative analysis. The FTIR spectrophotometer used a spectrum range of 400–4000 cm$^{-1}$ [27].

### 2.15. Differential Scanning Calorimetry (DSC)

A differential scanning calorimetry (DSC-50 system, Shimadzu, Kyoto, Japan) was used for the thermal analysis of spent grain powder. The analyses were performed in an atmosphere of nitrogen gas (99.99% purity), with the heating temperature ranging between 30–440 °C at a scan-rate of 10 °C/min and held for 1 min at 440 °C. The measurements were recorded and thermocouple-based temperature sensors were used for plotting the results [27].

### 2.16. X-ray Diffraction (XRD)

The crystalline structure, chemical bonding and affinity of particles of spent grain powder were studied by an analytical X-ray diffractometer (X′Pert PRO, PANalytical, Almelo, The Netherlands) to determine the diffraction XRD patterns using a Cu-based

anode X-ray tube. The wavelength of the X-ray was 1.5406 Å of Cu-Kα radiation and the analysis was conducted at a 5 °C/min scanning rate at 30 mA and 40 kV with a diffraction angle range of 5–50 (2θ) [27].

### 2.17. Total Plate Count (TPC)

The total plate count (TPC), known as aerobic plate count, was determined by the AOAC [28] method and calculated based on the equation:

$$\text{TPC (CFU/mL)} = \frac{\text{No. of colonies} \times \text{dilution factor}}{Volume\ of\ culture\ plate} \tag{15}$$

### 2.18. Total Fungal Count (TFC)

The total fungal count (TFC) was determined by AOAC [28] method and calculated using the given equation:

$$\text{TFC (CFU/mL)} = \frac{\text{No. of colonies} \times \text{dilution factor}}{Volume\ of\ culture\ plate} \tag{16}$$

### 2.19. Statistical Analysis

The standard error mean was calculated using Microsoft Excel 2019 (Microsoft Corp., Redmond, WA, USA). One-way (1-way ANOVA) and two-way analysis of variance (2-way ANOVA) confirmed the statistical difference in terms of significant and non-significant values, and the critical difference value was used to compare means. The results were recognized as statistically significant at $p \leq 0.05$.

## 3. Results and Discussion

### 3.1. Bulk Density (BD)

BD is the product′s complex property, which comprises particle density and the interstitial air present in the product. BD is influenced by several factors, with composition and particle size being two of the most significant. The result of the BD of spent grains is shown in Table 1. The lowest value was observed for BB (0.42 g/mL), whereas BF (0.63 g/mL) showed the highest value. All the samples—BB (0.42 g/mL), BM (0.45 g/mL), BW (0.50 g/mL), BR (0.56 g/mL) and BF (0.63 g/mL)—showed statistically significant differences ($p < 0.05$) from each other. Several studies reported that the BD of spent grains ranged between 0.14 and 0.35 g/mL [12], 0.6 g/mL [29] and 0.511–0.523 g/mL [30]. The BD of different grains was reported as being: 0.63–0.72 g/mL for barley grains [13], 0.718–0.805 g/mL for wheat [31], 0.644 g/mL for rice flour [32], and 0.30–0.48 g/mL for finger-millet [33]. The BD increases with the pressure and temperature but decreases by exceeding the critical point [12]. Additionally, if the spent grains are composed of various components with significantly different densities (e.g., husks, solids, and water), then this can impact the overall BD. Samples with finer particles tend to have higher tapped densities compared to samples with coarser particles, as they can pack more tightly. Spent grain samples with a higher fiber content may have lower tapped densities due to the presence of bulky, non-compactable material. The moisture content of the spent grain can significantly affect its tapped density as well. Understanding and controlling the BD of spent grains is important for various applications. Different industries and processes may have specific requirements for the BD of spent grains and adjustments in composition and particle size can be made to meet those requirements.

**Table 1.** Bulk density (BD), tapped density (TD), Carr′s index (CI), Hausner ratio (HR) and angle of repose (φ) (mean values ± standard deviation) of different samples of spent grain combinations.

| Samples | Bulk Density (g/mL) | Tapped Density (g/mL) | Carr′s Index | Hausner Ratio | Angle of Repose |
|---|---|---|---|---|---|
| BB | 0.42 ± 0.01 [a] | 0.48 ± 0.01 [a] | 12.50 ± 0.2 [c] | 1.143 ± 0.003 [c] | 21.16 ± 0.08 [a] |
| BW | 0.50 ± 0.02 [c] | 0.55 ± 0.02 [bc] | 9.00 ± 0.2 [a] | 1.100 ± 0.003 [a] | 21.80 ± 0.40 [b] |
| BR | 0.56 ± 0.01 [d] | 0.67 ± 0.03 [d] | 16.40 ± 0.3 [e] | 1.196 ± 0.004 [e] | 25.31 ± 0.03 [c] |
| BM | 0.45 ± 0.02 [b] | 0.53 ± 0.01 [b] | 15.00 ± 0.5 [d] | 1.178 ± 0.002 [d] | 32.09 ± 0.06 [e] |
| BF | 0.63 ± 0.01 [e] | 0.71 ± 0.02 [e] | 12.00 ± 0.3 [b] | 1.136 ± 0.005 [b] | 25.78 ± 0.02 [d] |

Note: Data are represented as mean ± standard deviation. [a–e] Mean values within a column with different superscript letters are statistically significantly different ($p \leq 0.05$) from each other. Spent grain combinations: barley (BB), barley and wheat (1:1, *w/w*) (BW), barley and rice (1:1, *w/w*) (BR), barley and maize (1:1, *w/w*) (BM), and barley and finger-millets (1:1) (BF).

*3.2. Tapped Density (TD)*

The value of TD shows variation in different samples. The result of the TD of spent grains is shown in Table 1. TD is an important physical property used to characterize the flowability and packing behavior of powders or granular materials. The lowest value of TD was observed for BB (0.48 g/mL), whereas the highest was for BF (0.71 g/mL). BW (0.55 g/mL) and BM (0.53 g/mL) showed non-significant differences ($p > 0.05$) from each other. In previous studies, the TD of spent grains was 0.18 to 0.38 g/mL [12]. The TD value for barley grains was 0.697–0.738 g/mL [13], for wheat was reported as 0.746–0.831 g/mL [31], for rice flour was 0.762 g/mL [32], and for finger-millet was 0.45–0.64 g/mL [33]. The TD increases with pressure and temperature, decreasing after approaching a critical value. Bulk and tapped density are important parameters for the powdered product with enhanced oxidative stability [12].

*3.3. Carr′s Index (CI)*

The result of the CI of spent grains is shown in Table 1. The lowest value of CI was found in BW spent grains (9.00%) and the highest value was seen in BR (16.40%). All the samples—BW (9.00%), BF (12.00%), BB (12.50%), BM (15.00%) and BR (16.40%)—showed statistically significant differences ($p < 0.05$) from each other. According to the literature, the CI of spent grains range between 12.3 and 35.3%, demonstrating the fine quality, fluidity and flowability of the product. Indeed, CI values lower than 15% show good flowability properties, while values above 25% show poor flowability properties [12]. The CI value was reported to be 3–11% for barley grains [13], 4.84–7.99% for wheat grain [31] and 15.48% for rice flour [32].

*3.4. Hausner Ratio (HR)*

The HR shows differences between samples of spent grains. The result of the HR of spent grains is shown in Table 1. The BR sample had the highest value of 1.196, with statistically significant differences ($p < 0.05$) observed among all samples, including BW (1.100), BF (1.136), BB (1.143) and BM (1.178). According to the literature, spent grains have an HR ranging from 1.09 to 1.56 [12], barley grains have an HR of 1.03–1.11 [13] and rice flour has an HR of 1.183 [32]. These HR values indicate good compressibility of the samples, making them suitable for food product development [27]. Additionally, the HR measures powder compressibility, and values greater than 1.35 indicate poor flow properties [34] and might suggest that the spent grains are more prone to clumping or sticking together, which could affect processing. A low HR might indicate that the spent grains flow easily and can be handled efficiently in the production process.

### 3.5. Angle of Repose (φ)

The angle of repose is the maximum angle at which a pile of granular material remains stable without sliding or collapsing due to gravity. The result of the angle of repose of spent grains is given in Table 1. The lowest and highest values were observed for BB (21.1°) and BM (32.09°), respectively. The angles of repose for BB (21.16°) and BW (21.80°) were significantly higher ($p < 0.05$) compared to BR (25.31°), BF (25.78°) and BW (21.80°). However, there was no statistically significant difference ($p > 0.05$) between the angles of repose for BR (25.31°) and BF (25.78°). According to the literature, the angle of repose of spent grain ranges between 23 and 36° [35], whereas for barley grains it ranges from 12 to 24° [13] and rice flour is 66.57° [32]. The angle of repose for spent grains can vary but it is generally in the range of 30 to 45°, depending on factors like moisture content and how compacted the grains are.

### 3.6. Water Absorption Index (WAI)

The WAI is a measure of the water-holding capacity of a material. It is commonly used in the food industry to assess the ability of a substance to absorb and retain water. In the context of spent grains, the WAI can be important for understanding their potential use in various applications. The result of the WAI of spent grains is tabulated in Table 2. The value of WAI for the control sample BB was observed to be 3.20 g/g. BM (5.03 g/g) showed a statistically significantly ($p < 0.05$) higher WAI when compared to BB (3.20 g/g), BF (3.56 g/g), BR (4.10 g/g) and BW (4.33 g/g). The WAIs of BB, BF, BR and BW (4.33 g/g) were not statistically different ($p > 0.05$). The literature reports a WAI range of spent grains between 1.3 and 1.52 g/g [36], 4.4 and 6.9 g/g [37] and 3.75 g/g [38]. This high WAI is attributed to the presence of hydroxyl groups in dietary fiber, which form hydrogen bonds with water [39,40]. The WAI of spent grains can vary depending on factors such as the type of grains used, the brewing process and how thoroughly they have been dried or processed after brewing. Generally, spent grains have a high water absorption capacity due to their fibrous nature. Carbohydrates and proteins also contribute to the WAI due to their hydrophilic properties, such as polar or charged side chains [41].

**Table 2.** Water absorption index (WAI), water solubility index (WSI), oil absorption capacity (OAC), swelling capacity (SC) and foaming capacity (FC) (mean values ± standard deviation) of different samples of spent grain combinations.

| Samples | WAI (g/g) | WSI (%) | OAC (g/g) | SC (mL/g) | FC (mL/g) |
|---|---|---|---|---|---|
| BB | 3.20 ± 0.53 [a] | 10.47 ± 0.41 [de] | 2.70 ± 0.05 [de] | 8.60 ± 0.30 [a] | 1.80 ± 0.20 [e] |
| BW | 4.33 ± 0.15 [cd] | 7.6 ± 0.69 [b] | 2.24 ± 0.04 [b] | 8.80 ± 0.10 [b] | 1.60 ± 0.10 [d] |
| BR | 4.10 ± 0.35 [c] | 8.20 ± 0.35 [abc] | 2.30 ± 0.30 [bc] | 8.80 ± 0.40 [b] | 1.00 ± 0.10 [b] |
| BM | 5.03 ± 0.41 [e] | 7.06 ± 0.31 [a] | 1.90 ± 0.40 [a] | 9.60 ± 0.20 [d] | 0.80 ± 0.10 [a] |
| BF | 3.56 ± 0.60 [b] | 9.67 ± 0.80 [cd] | 2.57 ± 0.03 [cd] | 9.00 ± 0.10 [bc] | 1.20 ± 0.20 [c] |

Note: Data are represented as mean ± standard deviation. [a–e] Mean values within a column with different superscript letters are statistically significantly different ($p \leq 0.05$) from each other. Spent grain combinations: barley (BB), barley and wheat (1:1, $w/w$) (BW), barley and rice (1:1, $w/w$) (BR), barley and maize (1:1, $w/w$) (BM), and barley and finger-millets (1:1) (BF).

### 3.7. Water Solubility Index (WSI)

The WSI is a measure of the solubility of a substance (usually a food ingredient) in water. It indicates the amount of material that can be dissolved in water under specified conditions. In the context of spent grains, the WSI can provide insights into their solubility characteristics. The results of the WSI of spent grains are displayed in Table 2. The WSI value for the control sample BB was reported to be 10.47%. BB (10.47%) and BF (9.67%) showed astatistically significantly ($p < 0.05$) higher WSIs when compared to BM (7.06%), BW (7.60%) and BR (8.20%). The WSIs of BM, BW and BR showed non-significant differences

($p > 0.05$) from each other. In previous studies, the WSI of spent grains ranged between 6.8 and 10.8% [37] and 13.69% [38]. The water solubility index is an indicator of the amount of water-soluble components present in the aqueous phase [39]. There is not a fixed or standard WSI for spent grains, as it can be different for each specific batch or type of grain. A higher WSI value is associated with increased stickiness and adhesiveness in food products [40].

### 3.8. Swelling Capacity (SC)

The results of the SC of spent grains is given in Table 2. The SC results of BB (8.60 mL/g), BW (8.80 mL/g) and BR (8.80 mL/g) showed statistically significantly ($p < 0.05$) lower values when compared with BF (9.00 mL/g) and BM (9.60 mL/g). In contrast, the SC of BB (8.60 mL/g), BW (8.80 mL/g) and BR (8.80 mL/g) presented no statistically significant variations ($p > 0.05$) from each other. In the literature, the swelling capacity of spent grains ranges from 13.00 to 14.00 mL/g [18] and 1.75 to 2.23 mL/g [42]. The SC allows the establishment of an enlargement rate of the particles due to water absorption and accumulation [18]. The SC depends on the particle size and the grain variety [43] and is an important factor to consider when using them for applications like food or animal feed. It affects their nutritional value, handling characteristics and suitability for various processes. The chemical composition of the spent grains, including residual sugars, proteins and fibers, can influence their swelling capacity.

### 3.9. Foaming Capacity (FC)

Different grains have varying protein and carbohydrate profiles, enzymatic activity and moisture content, which affect foaming properties. The result of the FC of spent grains is shown in Table 2. The lowest and highest values of FC were observed for BM (0.80 mL/g) and BB (1.80 mL/g), respectively. All the samples—BM (0.80 mL/g), BR (1.00 mL/g), BF (1.20 mL/g), BW (1.60 mL/g) and BB (1.80 mL/g)—showed statistically significant differences ($p < 0.05$) from each other. A previous study reported that the FC range of spent grains was 1.2 to 1.8 [36]. The low FC of spent grains is due to the globular protein, which resists surface denaturation [18]. The FC depends on the distribution of air bubbles in the liquid and semi-solid phases [44].

### 3.10. Oil Absorption Capacity (OAC)

The OAC of spent grains can be a significant factor due to their porous and fibrous nature. They can act as a natural absorbent for oils and fats. However, the exact absorption capacity can vary widely. For example, barley and wheat have distinct structural characteristics that can influence their oil absorption capacity. Table 2 displays the OAC results of spent grains. BM had a statistically significantly lower OAC value (1.90 g/g) than BW, BR, BF and BB (2.24 g/g, 2.30 g/g, 2.57 g/g and 2.70 g/g, respectively), with a *p*-value lower than 0.05. In contrast, BW, BR, BF and BB did not show any statistically significant differences ($p > 0.05$) in their OAC values. A previous study reported the OAC ranges of spent grains to be between 2.27 and 3.14 g/g [36]. Additionally, Li et al. [45] reported the OAC of BSG protein to be 3.10 g/g. Particle size affects the OAC, with finer particles having a higher oil absorption capacity [18]. The OAC is an important property for enhancing mouthfeel and retaining flavor and is mainly influenced by proteins that contain both hydrophilic and hydrophobic components [41].

### 3.11. Antioxidant Activity

Table 3 tabulates the results of the percentage of inhibition of DPPH, total phenolic content and total flavonoid content of different samples of spent grain combinations. The DPPH assay gives an indication of the antioxidant potential of the spent grain extract. It is one of many ways to assess the bioactivity of natural products. Spent grains have gained attention due to their potential as a source of important bioactive compounds, including phenolic compounds and flavonoids.

**Table 3.** Percentage of inhibition of DPPH (2,2-diphenyl-1-picrylhydrazyl), total phenolic content (TPC) and total flavonoids compounds (TFC) (mean values ± standard deviation) of different samples of spent grain combinations.

| Samples | % Inhibition of DPPH | TPC (mg GAE/g) | TFC (mg QE/g) |
|---|---|---|---|
| BB | 31.35 ± 0.81 [a] | 46.02 ± 0.26 [a] | 25.75 ± 0.72 [a] |
| BW | 36.61 ± 0.58 [b] | 55.87 ± 0.28 [d] | 38.67 ± 0.53 [bc] |
| BR | 39.11 ± 0.83 [c] | 50.22 ± 0.69 [b] | 35.88 ± 0.13 [b] |
| BM | 44.14 ± 0.56 [de] | 72.39 ± 0.61 [e] | 66.03 ± 0.87 [e] |
| BF | 43.03 ± 0.76 [d] | 50.94 ± 0.52 [bc] | 55.19 ± 0.67 [d] |

Note: Data are represented as mean ± standard deviation. [a–e] Mean values within a column with different superscript letters are statistically significantly different ($p \leq 0.05$) from each other. Spent grain combinations: barley (BB), barley and wheat (1:1, $w/w$) (BW), barley and rice (1:1, $w/w$) (BR), barley and maize (1:1, $w/w$) (BM), and barley and finger-millets (1:1) (BF). GAE = gallic acid equivalent; QE = quercetin.

The lowest and highest values of antioxidant activity were reported for BB (31.35%) and BM (44.14%), respectively. The samples BB, BW (36.61%), BR (39.11%), BF (43.03%) and BM all presented statistically significant differences ($p < 0.05$) from one another. The spent grains' scavenging activity was measured using the ascorbic acid standard curve (y = 0.0079x + 0.0484; $R^2$ = 0.9931). Based on the outcomes, it can be inferred that brewer's spent grain is a valuable source of antioxidants. In a study, the antioxidant activity of spent grains was reported to be in the range of 23 to 43% [46]. In two other studies, the % inhibition of DPPH for BSG ranged from 2.02 to 20.55 [47] and 5.33 to 17.18 [48]. The DPPH radical can react with the extract's hydrogen-donating species, like phenols and flavonoids [47].

*3.12. Total Phenolic Content (TPC)*

The TPC, expressed as mg GAE/g, of BR (50.229) and BF (50.94) showed a statistically significantly ($p < 0.05$) lower value when compared with BW (55.87) and BM (72.39). In contrast, the TPC content of BB (46.02), BW (55.87) and BM (72.391) showed no statistically significant variation ($p > 0.05$). The TPC content of spent grains was calculated using the standard curve of gallic acid (y= 0.0023x + 0.0255, R2 = 0.987). In a study, the TPC content of spent grains was reported to be 9.65 mg [7]. The phenolic content was reported to be in the range of 2.14 to 9.9 mg GAE/g [47]. The bound phenolic compounds are released during different stages of the malting process under the effect of temperature [49].

*3.13. Total Flavonoid Content (TFC)*

The TFC, expressed as mg QE/g, of BR (35.88) and BW (38.67), presented statistically significantly ($p < 0.05$) lower values when compared with BF (55.19) and BM (66.03). In contrast, the TFC of BB (25.75), BF (55.19) and BM (66.03) showed no statistically significant differences ($p > 0.05$) between each other. The TFC content of spent grains was calculated using the standard curve of quercetin (y= 0.0024x + 0.0002, R2 = 0.9996). In previous studies, the flavonoid content was reported to be in the range of 2.85 to 10.72 mg QE/g [49] and 8.80 to 50.38 mg QE/g [50]. The TPC and TFC content is influenced by the extraction techniques (solvent composition and extraction temperature and time) as well as factors like barley cultivar and the presence or absence of hull [49].

Understanding the phenolic and flavonoid content helps to evaluate the potential nutritional benefits of incorporating spent grains into food products or animal feed. If spent grains are found to be rich in phenolic compounds and flavonoids, they could potentially be used in the development of functional foods or dietary supplements. Naturally, utilizing spent grains for their phenolic and flavonoid content can be a sustainable approach to reducing waste in the brewing industry.

### 3.14. Amylase Inhibition Assay

Table 4 represents the results of the amylase and lipase inhibition assays of different samples of spent grains. The result of the amylase inhibition assay for the control sample BB was found to be 30.33%. The lowest value was reported for BW (10.22%), whereas BM (89.05%) showed the highest value. The samples BW, BB, BR, BF and BM displayed statistically significant differences ($p < 0.05$) from each other, with percentages of 10.22, 30.33, 37.02, 58.55 and 89.05, respectively. Previous studies have reported amylase inhibition assay values for unhydrolyzed and hydrolyzed BSG within the ranges of 8.08% to 13.35% and 15.5% to 24.4%, respectively, indicating low amylase activities in BSG and its hydrolysates. [51,52]. In another study, the $IC_{50}$ (half-maximal inhibitory concentration) value obtained in the BSG extracts (extraction with 60% acetone) was 35.5 [53]. Cian et al., 2018 [54] reported that digested BSG hydrolysates showed amylase inhibitory activity. According to another study, the amylase inhibition in BSG samples ranged from 15.5 to 24.4% [52].

**Table 4.** Amylase and lipase inhibition assay (mean values ± standard deviation) of different samples of spent grain combinations.

| Sample Code | % Inhibition of Amylase | $IC_{50}$ | % Inhibition of Lipase | $IC_{50}$ |
|---|---|---|---|---|
| BB | 30.33 ± 0.07 [b] | 8.58 | 24.57 ± 0.04 [b] | 6.41 |
| BW | 10.22 ± 0.06 [a] | 2.29 | 19.27 ± 0.03 [a] | 4.62 |
| BR | 37.02 ± 0.05 [c] | 11.90 | 44.51 ± 0.06 [c] | 24.46 |
| BM | 89.05 ± 0.09 [e] | 73.44 | 75.82 ± 0.07 [e] | 62.34 |
| BF | 58.55 ± 0.06 [d] | 27.02 | 61.26 ± 0.06 [d] | 32.49 |

Note: Data are represented as mean ± standard deviation. [a–e] Mean values within a column with different superscript letters are statistically significantly different ($p \leq 0.05$) from each other. Spent grain combinations: barley (BB), barley and wheat (1:1, *w/w*) (BW), barley and rice (1:1, *w/w*) (BR), barley and maize (1:1, *w/w*) (BM), and barley and finger-millets (1:1) (BF). $IC_{50}$ = Half-maximal inhibitory concentration.

### 3.15. Lipase Inhibition Assay

The result of the lipase inhibition assay for the control sample BB was found to be (24.57%) (Table 4). The lowest value was observed for BW (19.27%) and the highest value for BM (75.82%). All the samples—BW (19.27%), BB (24.57%), BR (44.51%), BF (61.26%) and BM (75.82%)—showed statistically significant differences ($p < 0.05$) from each other. In a previous study, the lipase inhibition in BSG samples ranged from 35.2 to 46.1% [52].

Amylase and lipase inhibition assays are methods used to assess the potential of compounds or extracts to inhibit the activity of these enzymes. These assays are often employed in the field of nutrition to screen the potential anti-diabetic or anti-obesity agents, as both amylase and lipase play crucial roles in the digestion and absorption of carbohydrates and lipids, respectively.

### 3.16. Glucose Uptake Assay

Glucose intolerance can cause energy imbalance, leading to obesity and diabetes. The samples underwent analysis to determine their impact on high glucose levels. The results are presented in Figure 1a,b. The glucose level and glucose movement were obtained at different time intervals. At 240 min, the glucose level and glucose movement for the samples were, respectively: BB (2.4 mg/dL and 65%), BW (2 mg/dL and 53%), BR (1.8 mg/dL and 57%), BM (2.2 mg/dL and 62%) and BF (2.1 mg/dL and 61%). Samples of BB (65%), BM (62%) and BF (61%) showed a high retention of glucose in the dialysis membrane. Therefore, these samples (BB, BM and BF) have the property of quenching the glucose molecule and play a role in enhancing high glucose levels. Indeed, spent grains can have a beneficial effect on blood sugar levels because they help to regulate the rate at which glucose is released into the bloodstream. This can help in preventing rapid spikes in blood sugar, which is especially important for individuals with diabetes or those trying to manage their blood sugar levels. Due to their high fiber content, spent grains can slow down the absorption of glucose from other foods in the digestive system.

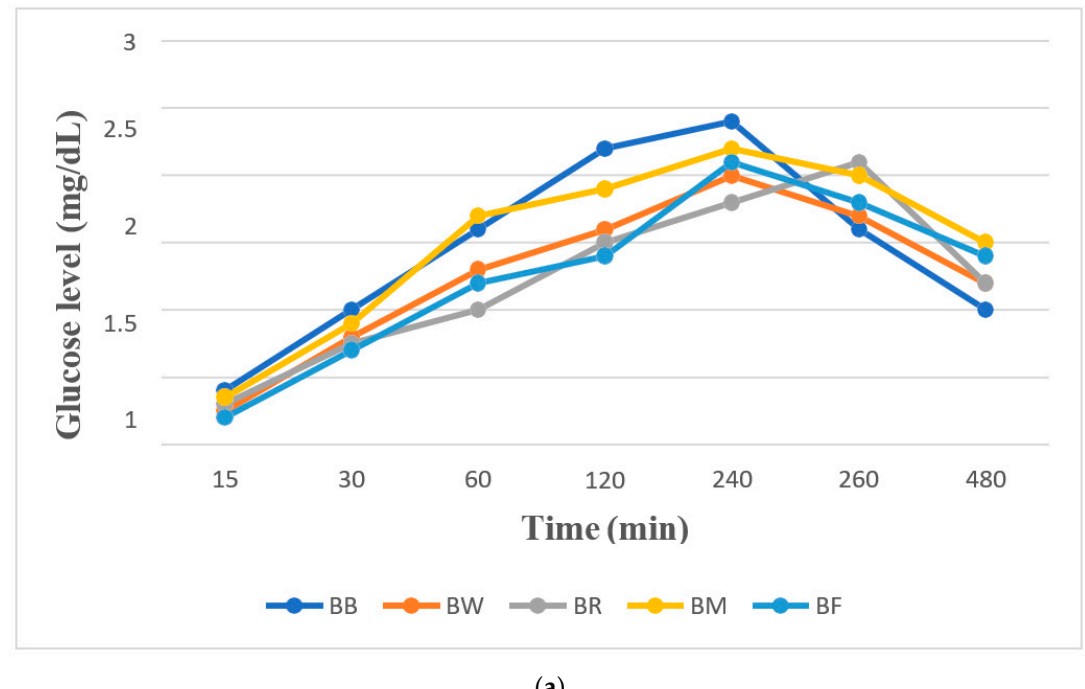

(**a**)

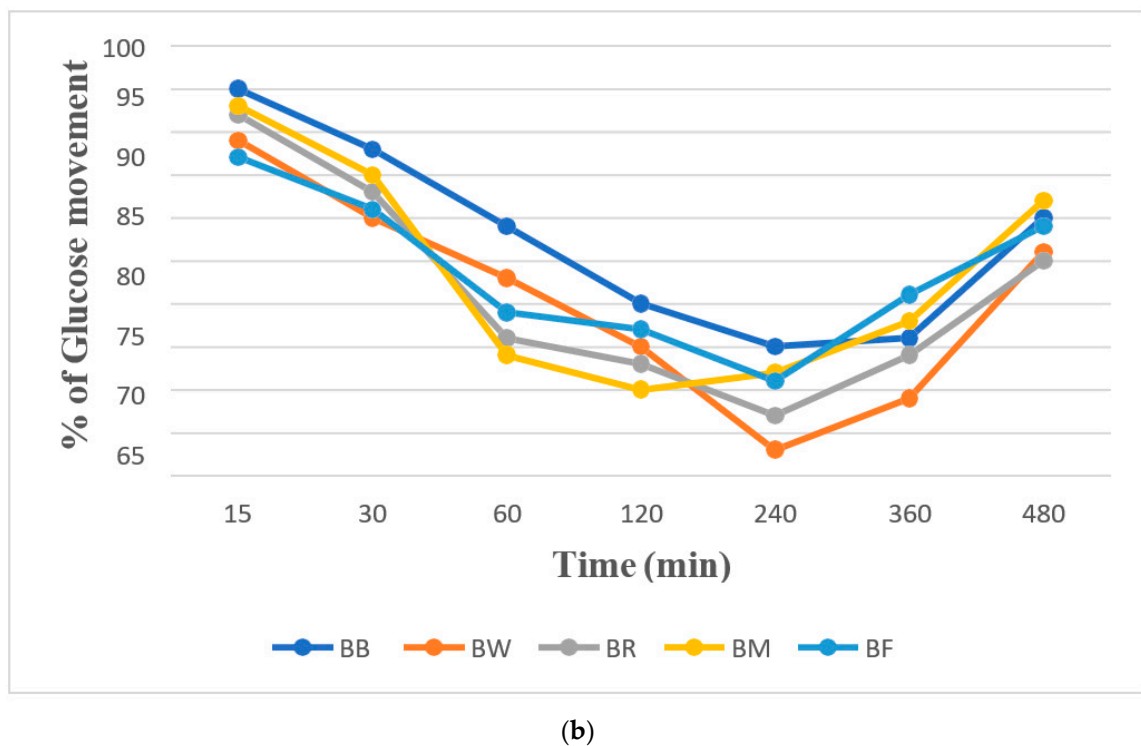

(**b**)

**Figure 1.** (**a**) Glucose level (mg/dL) and (**b**) percentage of glucose movement (observation values) of different samples of spent grain combinations: barley (BB), barley and wheat (1:1, *w/w*) (BW), barley and rice (1:1, *w/w*) (BR), barley and maize (1:1, *w/w*) (BM), and barley and finger-millets (1:1) (BF).

### 3.17. Total Protein Content (TPC)

Generally, spent grains contain around 20–30% of protein on a dry weight basis. The results of the TPC content of different samples of spent grain combinations are shown in Figure 2. The protein content for sample BB was found to be 21.14% (*w/w*). Similarly, BF (25.52% *w/w*) showed statistically significantly ($p < 0.05$) higher protein content compared with BB (21.14% *w/w*), BW (21.73%, *w/w*), BR (23.19%, *w/w*) and BM (23.63% *w/w*).

Moreover, the protein content of BB (21.14%, *w/w*), BW (21.73%, *w/w*), BR (23.19%, *w/w*) and BM (23.63%, *w/w*) showed no statistically significant differences ($p > 0.05$) between each other. These results were in accordance with previous studies, which reported that the protein content in brewer's spent grain was 19–30% (*w/w*) [9] and 22.13% (*w/w*) [55]. The protein content of BSG is a rich source of protein and it contains a high fraction of lysine, an essential amino acid, compared to other cereal products [9]. The human body cannot produce lysine on its own, so it must be obtained from dietary sources. This makes BSG a potentially valuable source of protein, especially for individuals who may have dietary restrictions or require higher lysine intake.

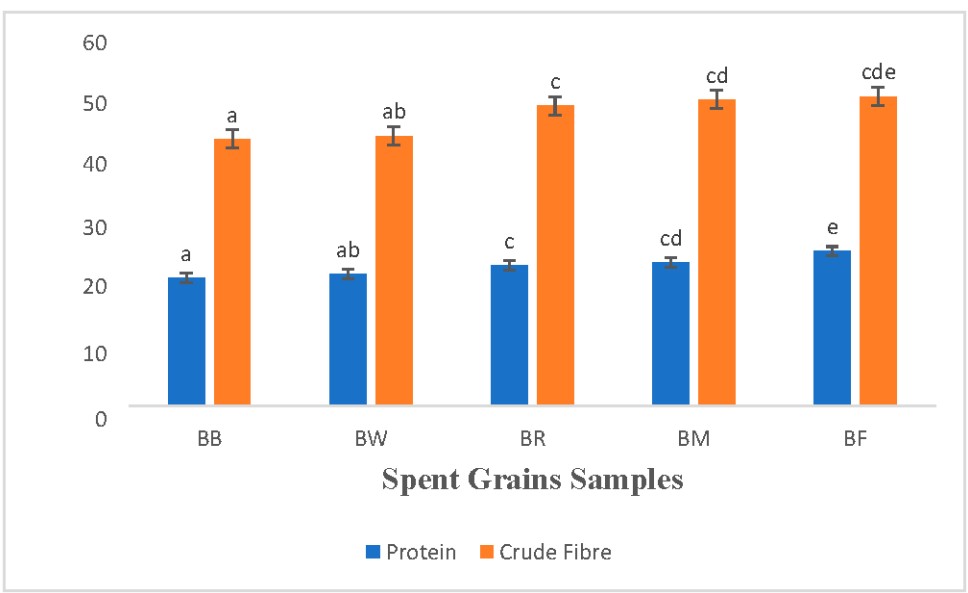

**Figure 2.** Total protein (TPC) and crude fiber (CF) contents (mean values ± standard deviation) of different samples of spent grain combinations: barley (BB), barley and wheat (1:1, *w/w*) (BW), barley and rice (1:1, *w/w*) (BR), barley and maize (1:1, *w/w*) (BM), and barley and finger-millets (1:1) (BF). [a–e] Means with the same superscript in a column do not vary significantly ($p < 0.05$) from each other.

*3.18. Crude Fiber (CF) Content*

Spent grains have a higher fiber content compared to the original grains because a significant portion of the carbohydrates and sugars have been extracted during the brewing process. The CF content of spent grains can range from approximately 25 to 50% on a dry matter basis. This is substantially higher than the CF content of the original grains before brewing. The results of the CF content of different samples of spent grain combinations are shown in Figure 2. The CF content for the sample BB was 44.00% (*w/w*). Likewise, BF (51.00%, *w/w*) showed a statistically significantly ($p < 0.05$) higher protein content when compared with BB (44.00, *w/w*), BW (44.50, *w/w*), BR (49.50, *w/w*) and BM (50.50%, *w/w*). Conversely, the crude fiber content of BB (44.00%, *w/w*) and BW (44.50%, *w/w*) showed statistically significant differences ($p < 0.05$) with BR (49.50%, *w/w*), BM (50.50%, *w/w*) and BF (51.00%, *w/w*). In previous studies, the CF content in BSG was reported to be 30–50 [9], 48.2 [55] and 41% (*w/w*) [47]. It is important to highlight that the high fiber and protein content in BSG make it suitable for food and non-food applications.

*3.19. Differential Scanning Calorimetry (DSC)*

The thermal properties of spent grains, such as denaturation temperature (Td) and enthalpy change (ΔH), as well as the identification of changes in a phase transition were identified using DSC. Denaturation temperature refers to the temperature at which the structure of a protein or enzyme starts to breakdown or unfold due to the disruption of non-covalent bonds (e.g., hydrogen bonds). The denaturation temperature of proteins in spent

grains can vary widely depending on the specific proteins present and their composition. For spent grains, the enthalpy change is relevant in processes like drying or roasting.

Figure 3 summarizes the DSC parameters for samples obtained at different temperatures ranging from 10–450 °C. The results of the sample BB revealed an endothermic peak at the onset temperature ($T_{onset}$) of 38.49 °C, end-set temperature ($T_{end-set}$) of 110.59 °C, denaturation peak temperature ($T_{peak}$) of 75.34 °C and variation of enthalpy of 99.6400 J/g. For the sample BW, the DSC parameters revealed an endothermic peak at a $T_{onset}$ of 42.44 °C, $T_{end-set}$ of 125.51 °C, denaturation $T_{peak}$ of 71.84 °C and $\Delta H$ of 56.5216 J/g. For the sample BR, an endothermic peak was observed at a $T_{onset}$ of 30.87 °C, $T_{end-set}$ of 104.52 °C, denaturation $T_{peak}$ of 73.48 °C and $\Delta H$ of 135.7388 J/g, and for the exothermic reaction, we observed a $T_{onset}$ of 297.31 °C, $T_{end-set}$ of 317.27 °C, denaturation $T_{peak}$ of 307.07 °C and $\Delta H$ 41.3196 J/g. For the sample BM, an endothermic peak was revealed at a $T_{onset}$ of 35.76 °C, $T_{end-set}$ of 111.94 °C, denaturation $T_{peak}$ of 73.26 °C and $\Delta H$ of 106.8021 J/g, and for the exothermic reaction, a $T_{onset}$ of 284.20 °C, $T_{end-set}$ of 324.92 °C, denaturation $T_{peak}$ of 305.20 °C and $\Delta H$ of 52.8977 J/g. Finally, for the sample BF, an endothermic peak at a $T_{onset}$ of 31.26 °C, $T_{end-set}$ of 129.58 °C, denaturation $T_{peak}$ of 70.93 °C and $\Delta H$ of 236.5729 J/g. The first onset of the peak is due to the release of the water molecule.

*3.20. X-ray Diffraction (XRD)*

In the context of spent grain samples, XRD analysis can be used to determine whether the grains exhibit a crystalline or amorphous structure. This information can be important for various applications, including understanding the nutritional content of the spent grains or for repurposing them in different industries. The components present in the biomass of spent grain samples can be amorphous or crystalline in nature. XRD analysis was used to determine the crystalline and amorphous nature of the samples, and the results are summarized in Figure 4. The diffraction pattern of the sample BB showed a substantial peak at 2θ values of 8.393°, 8.893°, 15.688°, 16.387°, 17.586°, 18.086°, 19.685°, 20.784° and 23.781°. For the sample BW, substantial peak values of 2θ were observed at 14.091°, 16.494°, 16.588°, 18.052°, 18.508°, 19.322° and 19.726°. For the sample BR, substantial peak values of 2θ were revealed at 16.080°, 19.043°, 20.640°, 20.695° and 22.191°. For the sample BM, substantial peak values of 2θ were observed at 14.357°, 16.333°, 18.392°, 18.573°, 18.727°, 20.282° and 20.450°. For the sample BF, substantial peak values of 2θ were revealed at 7.944°, 16.402°, 19.112°, 18.052°, 22.607° and 34.743°. The weak and unresolved peaks show that the sample is amorphous, while the sharp peak shows the crystallinity of the sample [56]. In the case of spent grains, if they have a crystalline structure, then it may suggest the presence of certain compounds or minerals in an ordered arrangement.

The spent grains predominantly showed an amorphous structure due to the presence of lignin and hemicellulose, while the presence of cellulose showed a crystalline structure [57]. The sample BF showed a sharp peak at 34.743°, which means that it has a high content of cellulose compared to lignin and hemicellulose, while all the other samples showed an amorphous structure due to weak and unresolved peaks. For the spent grains, an amorphous structure might indicate that the grains are primarily composed of non-crystalline materials, such as various organic compounds or glasses.

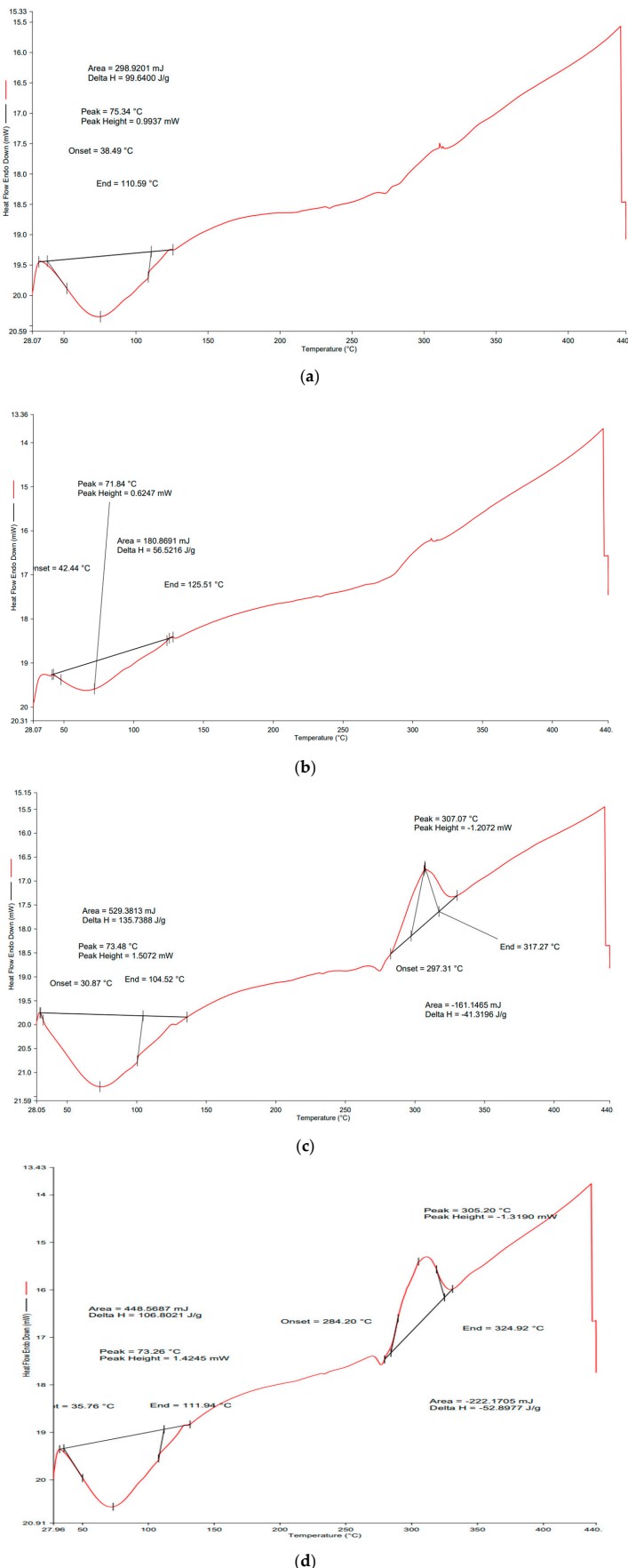

**Figure 3.** *Cont.*

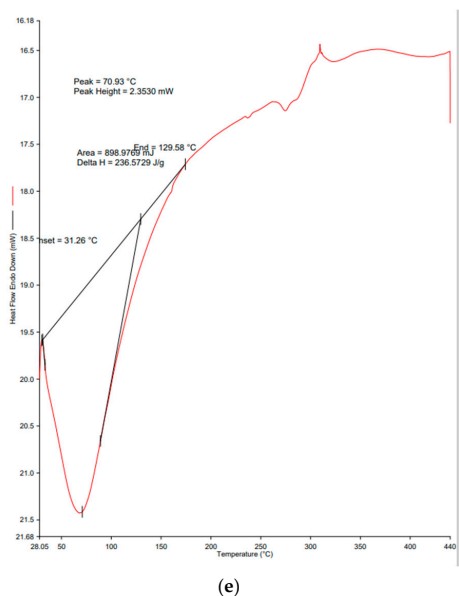

(**e**)

**Figure 3.** Differential scanning calorimetry (DSC) plots of different samples of spent grains: (**a**) barley (BB), (**b**) barley and wheat (1:1, $w/w$) (BW), (**c**) barley and rice (1:1, $w/w$) (BR), (**d**) barley and maize (1:1, $w/w$) (BM), and (**e**) barley and finger-millets (1:1) (BF).

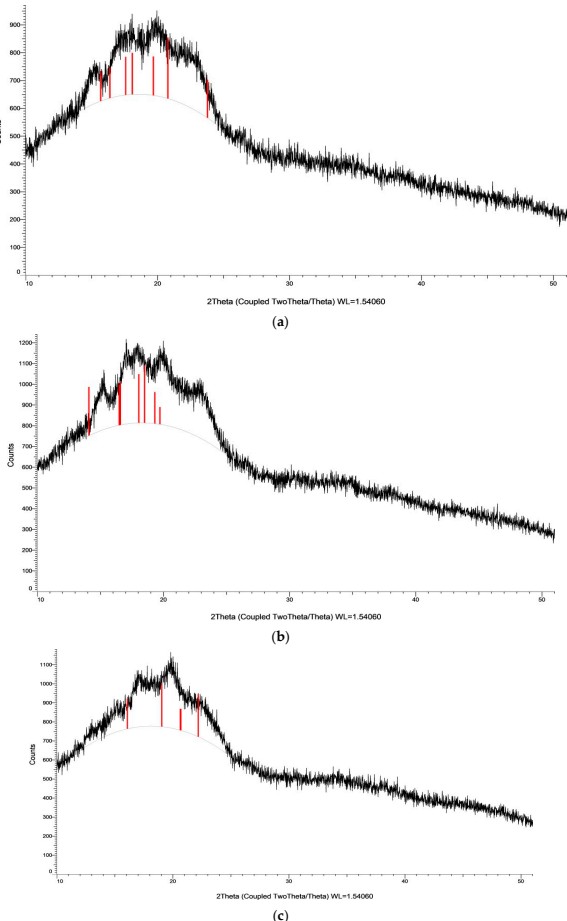

(**a**)

(**b**)

(**c**)

**Figure 4.** *Cont.*

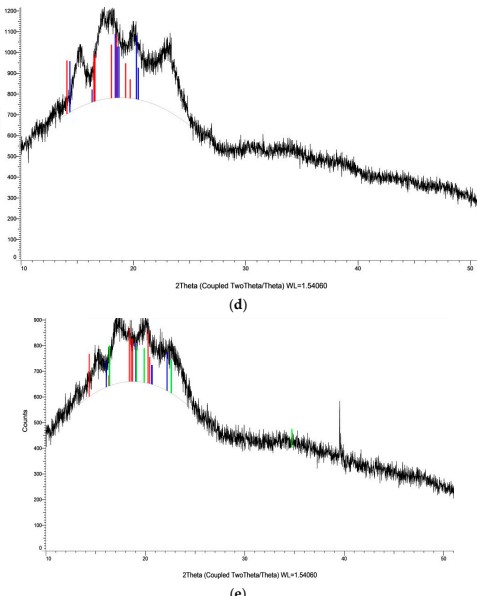

**Figure 4.** X-ray diffraction (XRD) plots of different samples of spent grains: (**a**) barley (BB), (**b**) barley and wheat (1:1, $w/w$) (BW), (**c**) barley and rice (1:1, $w/w$) (BR), (**d**) barley and maize (1:1, $w/w$) (BM), and (**e**) barley and finger-millets (1:1) (BF).

### 3.21. Fourier Transform Infrared Spectroscopy (FTIR)

Based on the functional properties, the samples were subjected to FTIR and recorded from 4000 to 400 cm$^{-1}$. Figure 5 represents the rotational and vibrational modes of different functional groups in different samples of spent grains. According to the FTIR spectra recorded, the sample BB revealed 9 peaks, detected at 3735.43, 2976.57, 2362.63, 1748.13, 1514.77, 1395.92, 1043.26, 669.06 and 520.81 cm$^{-1}$. The sample BW revealed 11 peaks, detected at 3285.35, 2362.41, 1744.15, 1645.90, 1517.49, 1396.09, 1242.46, 1016.39, 668.46, 570.28 and 530.13 cm$^{-1}$. The sample BR revealed 11 peaks, detected at 3288.79, 2924.43, 2362.43, 1646.97, 1516.00, 1396.26, 1147.75, 1011.37, 668.70, 570.98 and 521.93 cm$^{-1}$. The sample BM revealed 10 peaks, detected at 3281.65, 2923.11, 2362.62, 1744.47, 1647.96, 1515.88, 1396.05, 1012.44, 669.00 and 521.12 cm$^{-1}$. The sample BF revealed 11 peaks, detected at 3284.18, 2924.15, 2362.25, 1744.45, 1646.63, 1516.02, 1396.01, 1012.47, 668.75, 570.51 and 523.53 cm$^{-1}$.

The sample BB showed the stretching of various bonds at different wavelengths. O-H stretching vibrations of hydroxyl groups are caused by the intense peaks at 3735.43 cm$^{-1}$ [58]. Stretching in the hydroxyl group occurred at 3285.35, 3288.79, 3281.65 and 3284.18 cm$^{-1}$ in the samples BW, BM, BR and BF, respectively [59–61]. BB's strong C-H stretching vibration was observed at 2976.57 cm$^{-1}$ [62]. BR, BM and BF showed CH$_2$ asymmetric stretching vibration at peaks 2924.43, 2923.11 and 2924.15 cm$^{-1}$, respectively [63]. The absorption bands appeared at the peak of 2362 cm$^{-1}$, showing the N-H stretching vibrational bond of C = N-H on the pterin ring in folic acid in all the samples [64]. Strong C = O stretching in the vibrational bond was suggested at the peaks of 1748.13, 1744.15, 1744.47 and 1744.45 cm$^{-1}$ in the he samples BB, BW, BM and BF, respectively [65]. BW, BR, BM and BF at peaks 1645.90, 1646.97, 1647.96 and 1646.63 cm$^{-1}$ showed C = C stretching of alkenes [66]. The peaks at 1514.77, 1517.49, 1516.00, 1515.88 and 1516.02 cm$^{-1}$ showed the stretching vibrations of the C = C bond in all the samples [67,68]. All samples showed C = O symmetric stretching of COO−, which was assigned to the lipids at peaks 1395.92, 1396.09, 1396.26, 1396.05 and 1396.01 cm$^{-1}$ [69]. Strong stretching in the C-O vibration bond appeared at peaks 1242.46 and 1147.75 cm$^{-1}$ in the samples BW and BR, respectively. The absorption band of phosphate ion PO$_4^{3-}$ was observed at a frequency of 1043.26 cm$^{-1}$ in BB [70]. Stretching in the C-O-C vibrational bond was shown at peaks 1016.39, 1011.37, 1012.44 and 1012.47 cm$^{-1}$ in the samples BW, BR, BM and BF, respectively. Strong stretching in the C-Cl vibrational

bond was observed at peaks 668 and 669 cm$^{-1}$ in all the samples. Each sample showed strong stretching in the vibration bonds of C-Br at peaks between 500–600 cm$^{-1}$.

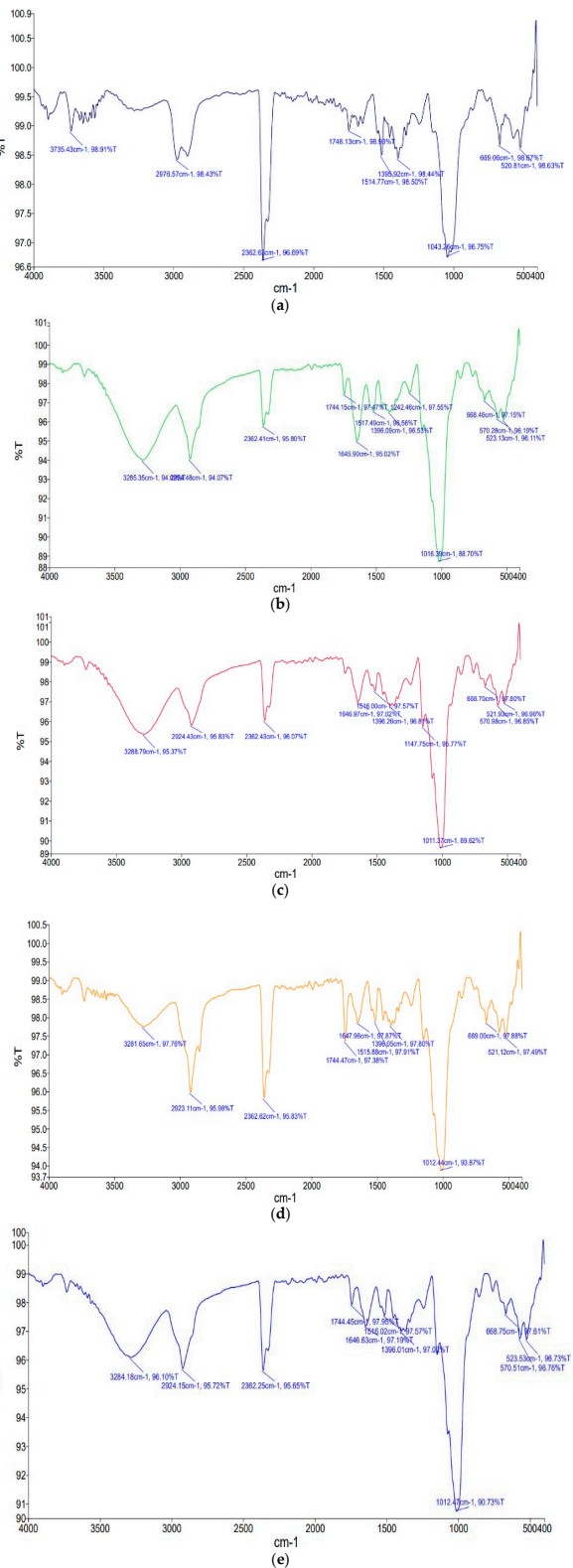

**Figure 5.** Fourier Transform Infrared Spectroscopy (FTIR) plots of different samples of spent grains: (**a**) barley (BB), (**b**) barley and wheat (1:1, *w/w*) (BW), (**c**) barley and rice (1:1, *w/w*) (BR), (**d**) barley and maize (1:1, *w/w*) (BM), and (**e**) barley and finger-millets (1:1) (BF).

Using FTIR on spent grains can be a valuable analytical tool. It can provide information about the chemical composition of the spent grains, including the presence of various organic compounds, proteins, lipids, carbohydrates and other molecules. This information can be useful for a variety of purposes, such as assessing the nutritional content of the spent grains, exploring potential uses for them (e.g., in animal feed, as a substrate for biofuel production or in food processing) or evaluating their environmental impact and potential for waste reduction or recycling.

*3.22. Total Plate Count (TPC)*

The TPC viable counts for the different sample were BB {1.16 log ((colony-forming units (CFU)/mL)}, BW (3 log (CFU/mL)), BR (1.63 log (CFU/mL)), BM (1.84 log (CFU/mL)) and BF (2.01 log (CFU/mL)). In previous studies, the range of microbial viable counts varied from $7 \times 105$ to $57 \times 105$ CFU/mL in spent grains in different concentrations and culture media [71].

*3.23. Total Fungal Count (TFC)*

The observed TFCs for the spent grain samples were BB (1.46 log (CFU/mL)), BW (1.86 log (CFU/mL)), BR (1.33 log (CFU/mL)), BM (1.54 log (CFU/mL)) and BF (2.040 log (CFU/mL)). The fungal viable counts of spent grains recorded were higher than 104 CFU/g [72], and the values ranged from 1.8–5 log (CFU/g) [73]. TFC for the spent grains can vary widely depending on factors such as the initial grain quality, the conditions in which it was processed and stored, and the specific fungal species present.

## 4. Future Outlook

Spent grains are a by-product of the brewing process and can contain various bioactive compounds. Working on the formulation of spent grains to enhance their therapeutic properties and incorporating them into various products with health benefits is an exciting area of research and development. Spent grains, the by-product of brewing processes, are typically rich in fibers, proteins, minerals and bioactive compounds. By optimizing their formulation, these grains can potentially offer higher therapeutic and nutraceutical properties against various harmful diseases. Experiments with different formulation techniques to enhance the therapeutic properties of spent grains are being performed based on the desired health benefits. This may involve incorporating other natural ingredients, such as herbs, spices, fruits or vegetables, with complementary health-promoting properties. The formulation should aim to maximize the bioavailability and stability of the bioactive compounds. The choice of application depends on the target audience, market demand and the specific targeted health benefits.

As with any by-product or waste material, appropriate handling, storage and processing are critical to ensuring safety and maximizing their potential benefits in various applications. Different compounds have different inhibitory potentials, and the concentrations of these compounds in the samples can also significantly influence the results of analysis. Additionally, local regulations and guidelines exist regarding the use of spent grains, particularly when using them as animal feed or human food products. Conducting sensory evaluations to ensure that the formulated spent grain products are not only nutritionally beneficial but also palatable and appealing to consumers is of foremost importance. Taste, texture, aroma and overall sensory experience play a crucial role in consumer acceptance and adoption of new and innovative products. Evaluating the microbiological quality given the rising consumption of edible products is of high importance, and outbreaks have been documented recently. Furthermore, the European Commission (EC) and different regional and national bodies mandate that food commodities should not contain pathogenic bacteria that pose dangers to public health [74–76].

## 5. Conclusions

Researching how to enhance the therapeutic properties of spent grains and using them in various health-benefiting products can have significant positive impacts on both human

health and waste management. In this study, we screened and analyzed different samples of spent grain combinations—viz., barley (BB), barley and wheat (1:1, $w/w$) (BW), barley and rice (1:1, $w/w$) (BR), barley and maize (1:1, $w/w$) (BM), and barley and finger-millets (1:1) (BF)—to determine their functional properties. BM had higher water absorption and swelling capacity, lower oil absorption and higher levels of DPPH inhibition, phenolic and flavonoid contents compared to BB, BW, BR and BF. Based on in vitro assays, BM also showed greater anti-obesity and anti-diabetic properties. Further work is necessary to develop spent grain formulations with even higher therapeutic properties against various harmful diseases and to incorporate them into different product applications with beneficial health effects.

Spent grains play a significant role in sustainability, especially within the context of brewing and distilling industries. Incorporating spent grains into different applications creates a closed-loop system where waste from one industry (brewing/distilling) becomes a valuable resource to another. This exemplifies the principles of a sustainable circular economy, where resources are reused and recycled to minimize waste and environmental impacts. There are clear sustainability benefits to repurposing spent grains, and it is important to ensure that the grains are used in a safe and responsible manner, as shown in the present research study.

**Author Contributions:** Conceptualization, M.K.; methodology, M.K.; writing—original draft preparation, M.K., A.A., D.K. and S.S.; formal analysis, A.A. and M.K.; writing—review and editing, M.K., D.K., A.V.R., J.K., M.T. and J.M.R.; visualization, M.K. and M.T.; supervision, M.K. All authors have read and agreed to the published version of the manuscript.

**Funding:** Author J.M.R. acknowledges the support made by LA/P/0045/2020 (ALiCE) and UIDB/00511/2020-UIDP/00511/2020 (LE-PABE) funded by national funds through FCT/MCTES (PIDDAC).

**Institutional Review Board Statement:** In this article, no ethical review and approval required due to already published data.

**Informed Consent Statement:** No requirement of consent as already published data were used.

**Data Availability Statement:** Not applicable.

**Acknowledgments:** The authors would like to acknowledge the COST Action 18101 SOURDOMICS—Sourdough biotechnology network towards novel, healthier and sustainable food and bioprocesses (https://sourdomics.com/; https://www.cost.eu/actions/CA18101/, accessed on 19 July 2023), where the author A.V.R. and M.T. are members of the working groups 1, 2, 6, 7, 8 and 9, and the author J.M.R. is the Chair and Grant Holder Scientific Representative and is supported by COST (European Co-operation in Science and Technology) (https://www.cost.eu/, accessed on 19 July 2023). COST is a funding agency for research and innovation networks. Author J.M.R. also acknowledges the Universidade Católica Portuguesa, CBQF—Centro de Biotecnologia e Química Fina—Laboratório Associado, Escola Superior de Biotecnologia, Porto, Portugal, as well as the LEPABE—Laboratory for Process Engineering, Environment, Biotechnology and Energy, and ALiCE—Associate Laboratory in Chemical Engineering, from the Faculty of Engineering, University of Porto, Porto, Portugal.

**Conflicts of Interest:** The authors declare no conflict of interest.

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
