# Peer review of "Combinations of Spent Grains as Sources of Valuable Compounds with Highly Valuable Functional and Microbial Properties"

_sustainability, doi:10.3390/su152015184_

Round 1

Reviewer 1 Report

The article “Spent grains combinations as sources of valuable compounds with high valuable functional properties” investigates BSG properties for application as therapeutical or other high-value products. The article is in the scope of Sustainability and reports novel content, however, it should be improved before final publication. A particular effort should be made to improve the Results and Discussion (consider the following comments):

1.      Abstract: all the acronyms should be defined in the abstract.

2.      Lines 76-77: the verb should be “be used”. The same in line 85.

3.      Introduction: the possibility of using spent grain for biogas production in anaerobic digestion processes should also be briefly mentioned. Consider 10.1016/j.jclepro.2017.07.197 and 10.1016/j.jece.2019.103184 as relevant references.

4.      Lines 111-115, 190-191, 353: the sentences are grammatically incorrect.

5.      Lines 277-281 should be moved to Materials and Methods.

6.      Results and discussion: the section should be completely rearranged. Most of the description in the text is a repetition of what is already presented in the Tables. The section is hardly readable and the English quality is low.

7.      Results and discussion: I would cut at least 50% of the text in lines 277-636 (see comment 6) and improve the discussion with the literature and the “future considerations” instead.

8.      The quality of Fig. 3 is low.

1.      The English language is poor and should be refined. Native speaker revision is suggested.

Reviewer 2 Report

The work has novel and relevant information, but the authors get bogged down in detailing the results without deep discussion.

The authors include too much information that does not lead to powerful conclusions. For example, microbiological analyzes do not contribute to the objective of the work and make the results very extensive, I would eliminate them. It is suggested not to repeat results that are in tables in the text, as is the case with WSI, WAI and OAC. It is suggested to close each report of results with a discussion of the results and check with the literature. An in-depth discussion should be made of the results that are simply reported.   The program used for ANOVAs must be detailed, remove from the summary acronyms not defined as WAI

Reviewer 3 Report

The research topic is interesting, but I would not recommend publishing it in its current format due to several areas that require significant improvement. Here are some specific comments:

·         The scientific quality of the work described in the manuscript is not generally satisfactory.

·         Abbreviations should be written on the first appearance. Identify WAI, and  OAC in the abstract.

·         Many sentences are missing hyphens. It is advisable to add the necessary hyphens.

·         Several sentences contain a series of three or more words, phrases, or clauses. Consider inserting commas to separate these elements.

·         Some sentences include unnecessary instances of "of" and "to."

·         Many sentences lack a comma after the introductory phrase.

·         Numerous sentences are missing proper article usage.

·         Some sentences have unnecessary articles.

·         The authors should be mindful of their use of "was" and "were" in the sentences.

·         Each equation should have a number.

·         ml should be written as mL throughout the whole manuscript  

·         Methods should be written in detail

·         Line 348 Titel without a table?

·         All tables need to be reformat

·         How did the authors identify the super-script litters, and which method was used(Tukey's multiple range test or Duncan's multiple ranges

·         The grammar and English throughout the manuscript need comprehensive revision. It is suggested to ask a native speaker to polish it.

The research topic is interesting, but I would not recommend publishing it in its current format due to several areas that require significant improvement. Here are some specific comments:

·         The scientific quality of the work described in the manuscript is not generally satisfactory.

·         Abbreviations should be written on the first appearance. Identify WAI, and  OAC in the abstract.

·         Many sentences are missing hyphens. It is advisable to add the necessary hyphens.

·         Several sentences contain a series of three or more words, phrases, or clauses. Consider inserting commas to separate these elements.

·         Some sentences include unnecessary instances of "of" and "to."

·         Many sentences lack a comma after the introductory phrase.

·         Numerous sentences are missing proper article usage.

·         Some sentences have unnecessary articles.

·         The authors should be mindful of their use of "was" and "were" in the sentences.

·         Each equation should have a number.

·         ml should be written as mL throughout the whole manuscript  

·         Methods should be written in detail

·         Line 348 Titel without a table?

·         All tables need to be reformat

·         How did the authors identify the super-script litters, and which method was used(Tukey's multiple range test or Duncan's multiple ranges

·         The grammar and English throughout the manuscript need comprehensive revision. It is suggested to ask a native speaker to polish it.

Reviewer 4 Report

The idea of the manuscript is very good and illustrated well. Utilization of food byproducts is very important for sustainable development and also for reduction of environmental pollution.

Dear authors please make all changes in the attached PDF file.

Round 2

Reviewer 2 Report

Comments, were included in the new manuscript.

Author Response

Dear Professor:

Once again, we would like to thank you  for your time spent on reviewing our manuscript entitled “Spent grains combinations as sources of valuable compounds with high valuable functional properties, and your insightful comments helping us improving the work. 

All changes were addressed and are with track changes record and also indicated by yellow shadow.

Sincerely yours,

The authors

Reviewer 3 Report

The authors have taken into consideration the most comments/suggestions of the reviewers during the revision of the manuscript, but the following minor points should be corrected.

 ((Determination of total phenols content)) changes phenols to phenol

 Changes aluminium to aluminum

Bulk Density (BD)

(( ISeveral studies reported )) Changes ISeveral to Several

Carr's Index (CI)

Adds the before literature

Hausner Ratio (HR)

((According to literature)) adds the before literature

Water Solubility Index (WSI)

((In the previous studies, the According to reports, the))

Remove s the before According

3.10.    Oil Absorption Capacity (OAC)

Removes the comma after flavor

3.12.    Total Phenolic Content (TPC)

Adds was before reported to be

3.13.    Total Flavonoid Content (TFC)

((between 2.85 mg)) Changes between to of

Changes occured to occurred

Page 23 removes although from (( Furthermore, although the European))

The authors have taken into consideration the most comments/suggestions of the reviewers during the revision of the manuscript, but the following minor points should be corrected.

 ((Determination of total phenols content)) changes phenols to phenol

 Changes aluminium to aluminum

Bulk Density (BD)

(( ISeveral studies reported )) Changes ISeveral to Several

Carr's Index (CI)

Adds the before literature

Hausner Ratio (HR)

((According to literature)) adds the before literature

Water Solubility Index (WSI)

((In the previous studies, the According to reports, the))

Remove s the before According

3.10.    Oil Absorption Capacity (OAC)

Removes the comma after flavor

3.12.    Total Phenolic Content (TPC)

Adds was before reported to be

3.13.    Total Flavonoid Content (TFC)

((between 2.85 mg)) Changes between to of

Changes occured to occurred

Page 23 removes although from (( Furthermore, although the European))

Author Response

(The authors gave the same response as above.)
